# Intranasal Prime–Boost with Spike Vectors Generates Antibody and T-Cell Responses at the Site of SARS-CoV-2 Infection

**DOI:** 10.3390/vaccines12101191

**Published:** 2024-10-18

**Authors:** Muriel Metko, Jason Tonne, Alexa Veliz Rios, Jill Thompson, Haley Mudrick, David Masopust, Rosa Maria Diaz, Michael A. Barry, Richard G. Vile

**Affiliations:** 1Department of Molecular Medicine, Mayo Clinic, Rochester, MN 55905, USA; metko.muriel@mayo.edu (M.M.); tonne.jason@mayo.edu (J.T.); velizrios.alexa@mayo.edu (A.V.R.); thompson.jillm@mayo.edu (J.T.); diaz.rosa@mayo.edu (R.M.D.); 2Molecular Pharmacology and Experimental Therapeutics Program, Mayo Clinic, Rochester, MN 55905, USA; mudrick.haley@mayo.edu; 3Department of Microbiology & Immunology, University of Minnesota Medical School, 2101 6th St. SE, Minneapolis, MN 55455, USA; masopust@umn.edu; 4Department of Infectious Diseases, Mayo Clinic, Rochester, MN 55905, USA; 5Department of Immunology, Mayo Clinic, Rochester, MN 55905, USA

**Keywords:** intranasal vaccine, SARS-CoV-2, viral vector, prime–boost

## Abstract

Background: Long-lived, re-activatable immunity to SARS-CoV-2 and its emerging variants will rely on T cells recognizing conserved regions of viral proteins across strains. Heterologous prime–boost regimens can elicit elevated levels of circulating CD8+ T cells that provide a reservoir of first responders upon viral infection. Although most vaccines are currently delivered intramuscularly (IM), the initial site of infection is the nasal cavity. Methods: Here, we tested the hypothesis that a heterologous prime and boost vaccine regimen delivered intranasally (IN) will generate improved immune responses locally at the site of virus infection compared to intramuscular vaccine/booster regimens. Results: In a transgenic human ACE2 murine model, both a Spike-expressing single-cycle adenovirus (SC-Ad) and an IFNß safety-enhanced replication-competent Vesicular Stomatitis Virus (VSV) platform generated anti-Spike antibody and T-cell responses that diminished with age. Although SC-Ad-Spike boosted a prime with VSV-Spike-mIFNß, SC-Ad-Spike alone induced maximal levels of IgG, IgA, and CD8+ T-cell responses. Conclusions: There were significant differences in T-cell responses in spleens compared to lungs, and the intranasal boost was significantly superior to the intramuscular boost in generating sentinel immune effectors at the site of the virus encounter in the lungs. These data show that serious consideration should be given to intranasal boosting with anti-SARS-CoV-2 vaccines.

## 1. Introduction

A critical problem with current vaccines against severe acute respiratory syndrome coronavirus 2 (SARS-CoV-2) is the emergence of viral variants that evade neutralizing antibodies (NAb), which also wane with time. Although most vaccines focus on potent NAb responses, the induction of long-lived, re-activatable immunity will rely upon strong T-cell responses against conserved regions of viral proteins across strains [1,2,3].

Heterologous prime–boost regimens with different immunogenic vectors can elicit extremely high levels of circulating CD8+ T cells against the shared target protein to provide a reservoir of first responders upon viral infection [4,5,6]. Furthermore, we, and others, have shown that mucosal immunization creates superior protection against mucosal infections compared to intramuscular or systemic immunization [7,8,9,10,11]. However, COVID-19 vaccines are currently delivered by intramuscular vaccinations, even though the initial site of virus infection is the nasal cavity. Most approved vaccines are replication-defective (RD) mRNA, DNA, or adenovirus (RD-Ad) vaccines [12]. Currently approved RD-Ads COVID-19 vaccines do not replicate transgenes or amplify antigen expression. In contrast, replication-competent adenovirus (RC-Ad) vaccines replicate the antigen gene up to 10,000-fold in the infected cell, producing 1–2 logs more immunogens per infected cell than their replication-defective counterparts but may also pose a real risk of causing viral infections as a side-effect of vaccination [13].

We developed a single-cycle Ad (SC-Ad) vector to allow for transgene DNA replication whilst avoiding the risk of Ad infections [9,14,15,16]. SC-Ad retains its E1 gene to allow genome replication but has the protein pIIIA deleted to prevent the production of infectious progeny viruses [9,14,15,16]. RC- and SC-Ad produce 33 to 100-fold more proteins than conventional RD-Ad [15]. However, SC-Ad does not generate infectious virions [16] and promotes a more robust and persistent immune response than RD- and RC-Ads [14,16,17].

Replicating VSV-based vaccines against several viral pathogens have been tested and even approved for use in humans [18,19,20]. Several VSV-based COVID-19 vaccine candidates have been generated by substituting the VSV glycoprotein with the Spike protein from SARS-CoV-2 [21,22,23]. This generates replication-competent, Spike-pseudotyped viruses that rely on the expression of the SARS-CoV-2 cellular ACE2 receptor on target cells for infection. A phase one clinical trial evaluating a VSV-Spike vaccine given IM showed suboptimal anti-viral humoral responses, likely due to the low expression of ACE2 at the vaccination site. In terms of mucosal immunity, IgG has antigen-specific antibody activity in the lower respiratory tract, and IgA prevents virus attachment to epithelial cells [24,25]. Previous studies have shown that the intranasal administration of a live attenuated SCD9 SARS-CoV-2 vaccine induces higher levels of IgA than intramuscular administration [26]. Similarly, an adenovirus-vectored SARS-CoV-2 vaccine, known as AdCOVID, which expressed the receptor-binding domain of the Spike protein, elicited a strong humoral and cellular response when administered intranasally [27]. Therefore, we hypothesize that a replication-competent, Spike-pseudotyped VSV-based vaccine platform would be more successful if administered IN because (1) ACE2 is expressed at high levels in the nasal mucosa and (2) the local intranasal induction of anti-Spike immune responses may be more effective in preventing an initial viral infection than an intramuscular vaccination. In this respect, we previously developed VSV-IFNß as an oncolytic platform [28,29,30], which expresses IFNß [29] to further repress its replication in normal cells [31,32,33], increase the immunogenicity of dying tumor cells and improve its clinical safety and efficacy [29,34,35].

In this study, we hypothesize that a heterologous prime and boost vaccine regimen SC-(Ad and VSV vectors) delivered intranasally (IN) will generate improved immune effector responses (both antibody and effector/memory T-cell responses) locally at the site of the virus infection compared to currently used intramuscular vaccine/booster regimens.

## 2. Materials and Methods

### 2.1. Cell Lines

HEK-293T cells engineered to over-express ACE2 (293T ACE2 cells) were provided by Dr. Paul Bieniasz (Laboratory of Retrovirology, The Rockefeller University, New York, NY, USA). These 293T ACE2 cells were selected with blasticidin at 0.5 μg/mL (#ant-bl-05, Invivogen, San Diego, CA, USA). BHK-21 [C-13] cells were originally obtained from the American Type Culture Collection (#CCL-10, ATCC, Manassas, VA, USA). Human A549 lung cells were provided by Dr. Michael Barry (Mayo Clinic, Rochester, MI, USA). Cells were grown in Dulbecco’s modified Eagle medium (DMEM; #SH30022.01 Hyclone, Logan, UT, USA) + 10% fetal bovine serum (FBS) (# A52567-01, Life Technologies, Carlsbad, CA, USA).

### 2.2. Viral Vector Design

We generated one SC-Ad-Spike and two VSV-Spike vectors that were replication-competent, wherein SARS-CoV-2 Spike, lacking its ER retention signal [31,36], was used to replace the VSV-G glycoprotein gene (Figure 1A). The SC-Ad vector expressed the Spike protein in the E4 region, whereas the VSV vectors expressed either the green fluorescent protein-luciferase (VSV-Spike-canto) or murine IFNß (VSV-Spike-mIFNß) genes upstream of the VSV L gene. The replacement of VSV-G with Spike makes VSV infections entirely contingent on binding to the ACE2 receptor of SARS-CoV-2, whereas the infection of target cells by the VSV-Spike vectors is mediated directly by the interaction of the Spike glycoprotein with the ACE2 receptor; SC-Ad-Spike expresses but does not use Spike as a cell-entry protein. Instead, it uses its fiber and penton base proteins for infection. SC-Ad and VSV viruses were validated for the expression of the Spike gene by Western Blot following the infection of 293T cells engineered to over-express ACE2 and of BHK cells, which naturally express ACE2 (Figure 1B), as well as for the mIFNß expression by ELISA (Figure 1C).

### 2.3. Vesicular Stomatitis Virus Pseudotyped with SARS-CoV-2 Spike

Vector Design: The Spike gene was isolated via a polymerase chain reaction from SARS-CoV-2 isolate Wuhan-Hu-1 (accession number NC_045512.2). The following primers (5′ SARS-F-Mlu1: TTTACGCGTCACTATGTTTGTCTTTCTCGTGCTG and 3′ SARS-CoV-2-delta-C-term TTTGCGGCCGCTTAGCAGCTTCCGCAGCTGCAGCA) were used to remove the ER retention signal of the native Spike protein [37,38] and create a Spike (delta 20) gene. Using MluI and NotI restriction sites, the SARS-CoV-2 Spike (delta 20) gene replaced the wild-type VSV-G glycoprotein gene in pVSV-XN2 or pVSV-XN2-GFPL to create pVSV-Spike plasmid and pVSV-Spike-GFPL plasmid, respectively. Then, the murine mIFNß gene was cloned into pVSV-Spike between the Spike and viral *L* gene using BsiWI and NheI to create pVSV-Spike-mIFNß plasmid.

Virus Rescue: VSV expressing SARS-CoV-2 Spike with green fluorescent protein-luciferase (VSV-Spike-GFPL) or mouse interferon beta (VSV-Spike-mIFNß) were rescued from pVSV-XN2 using reverse genetics in BHK cells, as described [29,33,36,39]. BHK cells in 6-well plates were infected with MVA-T7 for 1 h at 37 °C and 5% CO_2_. After 1 h, cells were transfected with either pVSV-Spike-mIFNß plasmid or pVSV-Spike-GFPL plasmid (2 µg), along with pBluescript (pBS, #212205, Agilent, Los Angeles, CA, USA) encoding VSV-N (1.1 µg), pBS encoding VSV-P (0.695 µg), pBS encoding VSV-L (0.6 µg), and pBS encoding VSV-G (0.55 µg) using Fugene6 (#284166, Promega, Madison, WI, USA), according to the manufacturer’s instructions. Forty-eight hours later, at full cytopathic effect, a culture supernatant was clarified by passing it through a 0.2 µm filter. Two rounds of plaque purification were performed using a noble agar overlay, using the protocol adapted from Schnell et al. [40] and Whelan et al. [39]. The virus was titered by TCID_50_, as described below.

### 2.4. Single-Cycle Adenovirus Expressing SARS-CoV-2 Spike

A codon-optimized cDNA encoding the Spike protein from the delta variant of the severe acute respiratory syndrome coronavirus 2 virus was inserted into a single-cycle adenovirus, as previously described [15]. The viruses were purified from polystyrene CellSTACK^®^—10 Chambers with Vent Caps (#3271, Corning, Charlotte, NC, USA)—on two CsCl gradients and were used as virus particles (vp) based on OD260 measurements [14,16,36].

### 2.5. Virus Titration by TCID_50_

The 50% tissue culture infectious dose (TCID_50_) is defined as the dilution of a virus required to infect 50% of a given batch of inoculated cell cultures [41]. 293T ACE2 cells were cultured in 5% FBS DMEM with blasticidin (0.5 μg/mL). Cells were diluted to 1.4 × 10^5^ cells/mL, and 50 μL were plated per well in a 96-well-plate format. Cells were incubated overnight at 37 °C, 5% CO_2_. Twenty-four hours later, VSV-Spike viruses were serially diluted at a 1:10 ratio. A total of 50 μL of each serial dilution was added to the plated cells, with 8 replicates per dilution, including 8 replicates of untreated cells as a negative control. Each virus titer was repeated in duplicate. After 72 h, cells were scored for the presence of a cytopathic effect (CPE). Wells with 30% or more CPE were considered positive. Titer was calculated based on the Spearman [42] and Kärber [43] methods. To estimate the PFU from TCID_50_, the Poisson distribution was applied; P(o) is the proportion of negative tubes, and ‘m’ is the mean number of infectious units per volume (PFU/mL), P(o) = e(−m). For any titer expressed as TCID_50_, P(o) = 0.5. Thus, e(−m) = 0.5 and m = −ln 0.5, which is ≈0.7. For example, one can assume that material with a TCID_50_ of 1 × 10^5^ TCID_50_/mL will produce approximately 0.7 × 10^5^ PFU/mL.

### 2.6. Western Blotting

293T ACE2 cells were infected with VSV-Spike-GFPL, VSV-Spike- mIFNß, or SC-Ad-Spike at multiplicities of infection (MOI) of 0.01 for VSV or 10,000 vp for SC-Ad, and they were harvested 24 h later. Cells were washed with PBS and treated with ice-cold a Radioimmunoprecipitation assay (RIPA) buffer (#R0278, Millipore Sigma, St. Louis, MO, USA) containing a protease inhibitor cocktail (10 µg/mL) to allow for cell lysis. The cell lysate was passed through a 21-gage needle and centrifuged at 13,000 rpm for 4 min at 4 °C, and the supernatant was collected. The protein was titrated by a Pierce BCA Protein Assay (#23228, Thermo FisherScientific, Ann Arbor, MI, USA). Laemmli’s denaturation buffer (#1610747, BioRad, Hercules, CA, USA) was added to the cell lysate and heat-treated at 95° for 10 min. The cell lysate was run by a Western Blot using an SDS-PAGE gel box using Mini-PROTEAN TGX 12% gel (#4561046, BioRad) at 120 V for 45 min. Chromatography paper was soaked in a transfer TGM buffer (tris, methanol, and glycine), and the polyvinylidene difluoride (PVDF) membrane (#IPVH304F0, Millipore Sigma) was rinsed with methanol. To complete the transfer, 5 sheets of chromatography paper, the PVDF membrane, the SDS-PAGE gel, and another 5 sheets of chromatography paper were stacked in a TransBlot^®^ Semi-Dry Transfer Cell (#1703940, BioRad) and run at 5.5 mAmp/cm^2^ for 25 min. After the transfer, the PVDF membrane was submerged in a blocking buffer (5% milk powder in TBST) for 40 min at RT on an orbital shaker. The membrane was incubated in a primary antibody and a SARS-CoV-2 (COVID-19) Spike antibody (1A9) (#GTX632604, GeneTex, Irvine, CA, USA) and diluted in a blocking buffer at 1:500 on an orbital shaker at 4 °C overnight. The membrane was washed for 15 min with Tris-buffered saline with 0.1% Tween^®^ 20 (TBST) detergent (#91414, Millipore Sigma) 3 times on an orbital shaker, incubated in a secondary antibody, Amersham ECL anti-mouse IgG HRP (#NA931V, GE Healthcare, Minneapolis, MN, USA) at 1:10,000 for 1 h at RT, washed for 15 min with 1X TBST 3 times on an orbital shaker, and coated with an ECL substrate and developed. The membrane was stripped with Restore^TM^ WB stripping buffer (#21059 Thermo Fisher Scientific, Waltham, MA, USA) and incubated for 15 min at 37 °C, rinsed with 1X TBST for 5 min, and incubated with anti-ß actin HRP (#A3854, Millipore Sigma) at 1:10,000 for 1 h. Finally, the membrane was washed for 15 min with 1X TBST 3 times on an orbital shaker, coated with an ECL substrate, and developed.

### 2.7. Anti-Viral Protein Assay

The LEGENDplex^TM^ Mouse Anti-Virus Panel 13 plex v.8 (#740622, Biolegend, San Diego, CA, USA) was used to assess the presence of viral proteins in homogenized brain tissue from mice treated with VSV-Spike-GFPL. This panel allows the simultaneous quantification of 13 mouse proteins, including IFN-γ, CXCL1 (KC), TNF-α, CCL2 (MCP-1), IL-12p70, CCL5 (RANTES), IL-1β, CXCL10 (IP-10), GM-CSF, IL-10, IFN-β, IFN-α, and IL-6. The assay was performed according to the manufacturer’s instructions and ran on a FACSCanto X SORP flow cytometer (BD Biosciences) with FACSDiva v8.0 software. Data were analyzed using the LEGENDplex software, and statistical analysis was performed using a one-way ANOVA.

### 2.8. In Vivo Studies

Female k18-hACEII mice were purchased from The Jackson Laboratory. All mice were maintained in Mayo Clinic’s pathogen-free BSL2 biohazard facility, accredited by AAALAC (Association for the Assessment and Accreditation of Laboratory Animal Care). All in vivo studies were approved by the Institutional Animal Care and Use Committee (IACUC) at Mayo Clinic.

For intramuscular (IM) immunization, 50 µL of the vaccine was injected into the mouse flank. For intranasal (IN) immunization, mice were anesthetized, and 25 µL of the vaccine was administered into each nostril for a total volume of 50 µL. The tip of a pipet tip was placed just inside the nostril to deliver the vaccine slowly.

### 2.9. Sample Collection

Serum: Blood was collected in blood collection tubes (Microtainer SST: #365967, BD, Franklin Lakes, NJ, USA) from the mice via a cheek bleed. The serum was isolated by allowing the blood to clot, centrifuging at 4000 rpm for 6 min, and pipetting the serum from the top layer. The mice were euthanized by CO_2_ inhalation at the study endpoint to collect spleens and bronchoalveolar (BAL) fluid.

BAL: The Mice were sprayed with 70% ethanol. Scissors were used to open the chest cavity and expose the trachea. An oral gavage needle was inserted into the trachea. The lungs were rinsed with 1 mL of PBS three times for a total volume of 3 mL. BAL was centrifuged to collect cells. BAL cells were treated with 200 µL of ACK for 2 min. The ACK buffer was comprised of 500 mL of nanopure H_2_O, 4.01 g of NH4Cl (#A4514-500G Millipore Sigma), 0.5 g of KHCO3 (#237205 Millipore Sigma), and 100 µL of 5 mM EDTA (#AM92606 Thermo Fisher Scientific). The cells were then neutralized with PBS and centrifuged at 1400 rpm for 4 min. Then, they were resuspended in PBS for flow cytometry. BAL fluid was used for ELISA.

Spleens: Spleens were removed from the body cavity and placed in RPMI (#10-040-CV, Corning) on ice. The spleens were smashed through a 100 μm filter with a plunger, rinsed with 10 mL of PBS, and centrifuged at 1400 rpm for 4 min. Cell pellets were treated with 2 mL of ACK for 2 min, neutralized with 12 mL of PBS, and centrifuged. The cells were resuspended in PBS for flow cytometry.

### 2.10. Anti-Spike Antibody ELISA

Binding IgG and IgA antibody responses in mouse serum and bronchoalveolar lavage fluid were measured by ELISA against the Spike S1 protein [14]. Nunc Immuno Maxisorp plates (#442404, Thermo Fisher Scientific) were coated with 100 ng/well of a Recombinant SARS-CoV-2 (2019-nCoV) Spike Protein S1 Subunit, Fc Tag (#40591-V02H), in 100 µL PBS. The plates were incubated at 4 °C overnight. The plates were washed twice with 200 µL 1X TBST followed by 200 µL of a blocking buffer (5% milk powder in TBST) for 2 h at room temperature (RT). The plates were washed twice with 200 µL 1X TBST. Serum and bronchoalveolar samples were diluted in a blocking buffer (1:1000), transferred to the assay plate, and incubated at RT for 3 h. The samples were run in triplicate, including a triplicate of positive and negative control wells. A SARS-CoV-2 Spike S1 subunit antibody (MAB105403, R&D systems, Minneapolis, MN, USA) was used as the positive control. Serum and bronchoalveolar samples from K18-hACE2 mice treated with PBS served as the negative control. The plates were left to incubate for 3 h at RT. The plates were washed 4 times with 200 µL 1X TBST. An anti-mouse IgG HRP antibody (#A16078, Thermo Fisher Scientific) and an anti-mouse IgA HRP antibody (#626720, Thermo Fisher Scientific) were used as the secondary antibodies at a 1:5000 dilution in the blocking buffer. The plates were left to incubate for 2 h at RT. The plates were washed 4 times with 200 µL 1X TBST. A total of 50 µL of the 1-Step Ultra TMB-ELISA substrate (optEIA, #555214, BD) was added to each well and incubated at RT for 30 min; then, 50 µL of 2M sulfuric acid was added to each well. The plates were read at 450 nm in an Infinite M200 Pro Tecan plate reader. Statistical analyses were performed by one-way ANOVA.

### 2.11. Flow Cytometry

Splenocytes and cells from the BAL were processed as described above. The 1 × 10^6^ splenocytes were plated per well. All BAL cells were plated due to their limited number. Zombie NIR^TM^ (#423105, Biolegend) was used at 1:1500 in PBS for live/dead staining. The following antibodies were used for extracellular staining: PE/Cyanine 7 CD8b.2 (clone 53-5.8, Biolegend, #140416), PE/Dazzle^TM^594 CD44 (clone IM7, Biolegend, #103056), APC CD8b.2 (clone 53-5.8, Biolegend, #140410), and BV421 CD8a (clone 53-6.7, Biolegend, #100738) (used at 1:1000). FITC CD44 (clone IM7, Biolegend, #103006) was used at 1:200. Additionally, tetramers from the NIH Tetramer Core Facility were used: APC-labeled H-2K(b) VNFNFGL tetramer for the SARS-CoV-2 Spike protein and BV421-labled H-2K(b) RGYVYQGL for the VSV-Nucleocapsid protein. The samples were run on a ZE5 Cell Analyzer (Bio-Rad) with Everest™ v2.4 software and analyzed on FlowJo v.10.7.2. Statistical analysis was performed in GraphPad Prism.

### 2.12. Statistical Analysis

All statistical analyses were completed in GraphPad Prism 9 using one-way ANOVA, a *t*-test, or the Grehan–Breslow–Wilcoxon test, depending on the experiment.

## 3. Results

### 3.1. Safety and Immunogenicity of Intranasal Delivery of Viral Vaccines Expressing SARS-CoV-2 Spike

We hypothesized that intranasal (IN) vaccinations would be important to generate robust T-cell responses to SARS-CoV-2 at the time of the initial viral infection. However, when VSV-Spike-GFPL without mIFNß was used intranasally, the vector was toxic to mice after five days of virus administration (Figure 1D). The brains of all mice that experienced lethal viral toxicity contained both detectable infectious VSV-Spike-GFPL viruses and evidence of a storm of inflammatory cytokines, neither of which was present in the uninfected control mice (Figure 1E). When VSV-Spike-GFPL was administered IN at the highest dose of 5 × 10^5^ pfu to wild-type C57Bl/6 mice (not expressing ACE2), no toxicity, infectious virus, or cytokine storm was detected. In contrast, VSV-Spike-mIFNß was safe at all doses tested, validating the use of mIFNß as a safety feature within the vaccine (Figure 1D). We, and others [29,33,34,44,45], have previously shown that the addition of IFNß to VSV significantly reduces the toxicity and cytokine storm induced by VSV alone. Based on these studies, we would expect the magnitude of the cytokine profile shown in Figure 1E to be significantly reduced both quantitatively and qualitatively. Further studies will be required to identify the exact cytokines/chemokines responsible for the acute toxicity we observed with VSV-GFPL in Figure 1D.

### 3.2. Intranasal, but Not Intramuscular, Delivery of Viruses Induces T-Cell Responses in the Lung Airways

To test our hypothesis that intranasal vaccination would produce higher levels of protective anti-Spike T-cell responses at the site of the initial infection, we measured both the humoral and cellular immune responses generated by VSV-Spike-mIFNß and SC-Ad-Spike 14 days after administration either IN or intramuscularly (IM) in hACE2-transgenic mice (Figure 2A). Both vectors by both routes of viral administration induced similar levels of anti-spike IgG antibodies in the blood (Figure 2B). The intranasal delivery of both VSV-Spike-mIFNß and SC-Ad-Spike induced an influx of CD8 T cells into the bronchoalveolar lavages (BALs), including anti-spike T cells (Figure 2C,D). In contrast, intramuscular administration did not generate detectable levels of anti-Spike T in BALs. Although an intranasal vaccination with VSV-Spike-mIFNß induced a clear trend of increased numbers of anti-Spike, tetramer-positive T cells, this did not reach a significant level over IM-treated mice (Figure 2D). Intranasal vaccination with SC-Ad-Spike showed a trend towards differences between IM-vaccinated mice and IN-VSV-Spike-mIFNß vaccinated mice (Figure 2D). Comparable levels of anti-spike CD8 T cells were detected in the spleen between the routes of vaccination with the same vector and between the vector types (Figure 2E). T-cell responses against the VSV nucleocapsid protein largely mirrored those against the VSV-encoded Spike protein in both bronchoalveolar lavage and spleens (Figure 2C–E).

### 3.3. SC-Ad-Spike Induces Anti-Spike IgA Response in a Homologous Prime and Boost Regimen

We hypothesized that a heterologous prime and boost regimen would significantly enhance the generation of anti-Spike T-cell responses (Figure 3A). Neither homologous nor heterologous prime and boost vaccinations improved after a single vaccination with either vector for the induction of anti-Spike IgG responses in either blood or bronchoalveolar lavage (Figure 3B,C). Similarly, homologous prime and boost with VSV-Spike-mIFNß did not improve the levels of anti-Spike IgA induced in the BAL compared to a single vaccination with VSV-Spike-mIFNß (Figure 3D). However, SC-Ad-Spike, either as the heterologous prime or boost, significantly enhanced levels of anti-Spike IgA in the BAL compared to a single vaccination with either VSV-Spike-mIFNß or SC-Ad-Spike (Figure 3D). Moreover, SC-Ad-Spike also generated significantly enhanced anti-Spike IgA when used as a homologous boost (Figure 3D). Considering that IgA is most abundant in the mucosal surface and protects the host by neutralizing pathogens and preventing them from attaching to epithelial cells, ref. [37] these enhanced anti-Spike IgA levels in the mucosal surfaces could have utility in limiting SARS-CoV-2 infections at its site of entry.

### 3.4. SC-Ad-Spike Induces Anti-Spike CD8+ T-Cell Responses in the Lungs Independent of Prime and Boost Regimens

A single vaccination with SC-Ad-Spike once again induced both significantly higher levels of CD8+ T cells and anti-Spike CD8 T cells in the BAL than VSV-Spike-mIFNß (Figure 3E). By day 42, 21 days after the boost vaccinations, homologous and heterologous prime and boost regimens generally enhanced the levels of anti-Spike CD8+ T cells in BALs compared to a single SC-Ad-Spike vaccination, although these changes were not significant (Figure 3E,F). Homologous prime and boost with VSV-Spike-mIFNß did not significantly enhance the low anti-Spike CD8+ T-cell responses in the BAL compared to a single VSV-Spike-mIFNß vaccination (Figure 3E,F). In contrast, homologous prime and boost with VSV-Spike-mIFNß did significantly boost the levels of anti-VSV-N CD8+ T cells (Figure 3E,F). This suggested that SARS-CoV-2 Spike may be less immunodominant than VSV-N in C57Bl/6 mice or that differing responses are generated vs intracellular antigens than antigens displayed on the surfaces of cells or virions.

Relevant to the order of heterologous prime–boost, a first vaccination with SC-Ad-Spike largely blocked the generation of anti-VSV-N CD8+ T cells (Figure 3E,F). These data suggest that an initial SC-Ad-Spike vaccination generates anti-Spike antibodies and CD8+ T cells that can prevent subsequent infection with the VSV-Spike-mIFNß boost (which relies on Spike-ACE2 receptor binding), thereby inhibiting the generation of anti-VSV CD8+ T-cell responses in this regimen.

Differences between T-cell responses in the spleen and BALs were once again observed. Heterologous prime: boost with VSV-Spike-mIFNß:SC-Ad-Spike significantly increased Spike-tetramer-positive CD8+ T cells in the spleen when compared to single vector vaccinations or homologous prime and boost with VSV-Spike-mIFNß or homologous or heterologous prime and boost with SC-Ad-Spike (Figure 3G).

### 3.5. Anti-Spike Ab and T-Cell Responses Decrease with the Age of the Host

Susceptibility to and the severity of SARS-CoV-2 infections in humans increase with age. Therefore, we repeated the experiments of Figure 3 in old (78 weeks) K18-hAce2 mice (Figure 4). Despite a trend toward a significant increase in IgG levels with boosting by several of the regimens, no significant increases were observed in these aged mice with any of the prime and boost regimens when they were compared to the priming vaccination alone (Figure 4B). In general serum antibody levels were lower in the old mice than in younger mice.

The Sc-Ad-Spike vector continued to generate the highest levels, either alone or with a homologous boost (Figure 4B). Similarly, a homologous prime and boost with SC-Ad-Spike:SC-Ad-Spike was the most effective at generating anti-Spike CD8+ T-cell responses in the BAL, significantly more so than a single VSV-Spike-mIFNß vaccine (Figure 4C). However, this homologous prime and boost with Sc-Ad-Spike was not significantly different from other single or prime and boost regimens in these aged mice (Figure 4C).

As with the young mice, VSV-Spike-mIFNß boosted the anti-VSV-N T-cell response following an initial prime with VSV-Spike-mIFNß, but previous treatment with SC-Ad-Spike significantly reduced the anti-VSV T-cell response, presumably due to the neutralization of the boost dose of VSV-Spike-mIFNß by SC-Ad-Spike-induced anti-Spike Ab, which prevented infection by the incoming VSV (Figure 4C). As for the young mice, anti-Spike T-cell responses in the spleen did not reflect differences seen in the BAL and were not significantly different between different vaccination regimens (Figure 4D).

### 3.6. In Boosting Significantly Enhances IgA and CD8+ T-Cell Responses Compared to Intramuscular Boosting

Most humans who have been immunized against SARS-CoV-2 have received an anti-SARS-CoV-2 vaccination by the intramuscular (IM) route. We, therefore, tested in SC-Ad-Spike IM-primed mice whether a subsequent intranasal boost (at the site of likely viral encounter) would increase antibody and T-cell responses (Figure 5A). Under these conditions, an intranasal boost with SC-Ad-Spike or VSV-Spike-mIFNß did not enhance the levels of anti-Spike IgG in the blood or in BALs when compared to the intramuscular SC-Ad-Spike prime (Figure 5B,C). In contrast, an intranasal boost with SC-Ad-Spike significantly increased levels of anti-Spike IgA in the BAL compared to priming alone with SC-Ad-Spike (Figure 5D), an effect which was also significantly better than an intranasal boost with VSV-Spike-mIFNß (Figure 5D). As before, intranasal treatment with SC-Ad-Spike (in the absence of a prime vaccine) was also significantly more effective than intranasal VSV-Spike-mIFNß in inducing anti-Spike IgA responses in the BAL (Figure 5D).

IN boost with SC-Ad-Spike following intramuscular vaccination with SC-Ad-Spike induced a highly significantly increased CD8+ infiltration into the BAL compared to no intranasal boost, a large proportion of which was comprised of anti-Spike CD8+ T cells (Figure 5E). An intranasal boost with VSV-Spike-mIFNß was unable to do the same (Figure 5E), and a single intranasal vaccine with SC-Ad-Spike was also significantly more effective than with VSV-Spike-mIFNß (Figure 5E). Once again, pre-vaccination (IM) with SC-Ad-Spike completely prevented the induction of anti-VSV-N T-cell responses following boost with VSV-Spike-mIFNß presumably due to the inhibition of VSV-Spike-mIFNß infection through the Spike-ACE2 interaction (Figure 5E). Interestingly, in the setting of intramuscular prime and intranasal boost, similar trends in the generation of anti-Spike and anti-VSV T-cell responses were seen in the spleen as in the BAL, suggesting that intranasal boosting can contribute significantly to systemic immune responses to Spike (Figure 5F).

Although intranasal boosting of the intramuscular SC-Ad-Spike prime was not significantly different from intramuscular boosting for the induction of anti-Spike IgG in the BAL (Figure 5G), only the intranasal boost was able to significantly raise both anti-Spike IgA and anti-Spike CD8+ T-cell levels compared to no boost or to an intramuscular boost (Figure 5H).

## 4. Discussion

Our primary goal was to test whether a heterologous prime and boost regimen would improve upon a single-vaccinating platform to generate anti-SARS-CoV-2 immune responses. Our second major goal was to investigate whether intranasal vaccination can generate robust T-cell responses against SARS-CoV-2 by recruiting sentinel immune effector mechanisms directly to airway epithelial barriers in the lungs where the first encounters of an infection by the virus will occur. We tested replication-competent VSV-Spike and SC-Ad-Spike based on the proven ability of VSV to serve as a vaccine platform for other viral diseases and the ability of SC-Ad to generate very high levels of immunogen expression.

We show here that the incorporation of the IFNß gene into the VSV-Spike vector significantly enhanced viral attenuation and safety for intranasal vaccination and that the toxicity of VSV-Spike (no mIFNß) required a Spike-mediated infection of ACE2-expressing cells (Figure 1). We believe that only the VSV-Spike-GFPL was toxic for a variety of reasons. In the first instance, this is a replication-competent virus that spreads in vivo through the binding of Spike to the ACE2 receptor, thereby generating progressively increasing amounts of the virus. VSV has well-documented neurotoxic properties, so the intranasal spread of this virus at sufficiently high titers will manifest as neurotoxicity [46]. In addition, viral spread and infection are associated with a cytokine release syndrome that has the potential to generate significant toxicity itself [47]. The addition of the IFNß gene to the VSV-Spike-IFNß virus largely abrogated this toxicity because high levels of Type I IFNs, such as IFNß, are highly inhibitory to VSV replication and spreading, hence restricting the direct neurotoxicity of VSV and the generation of the associated cytokine storm [48]. Finally, since the ScAd-Spike is not a replication-competent virus, the toxicities associated with the replicative spread of the VSV-Spike would not be induced.

Both VSV-Spike-mIFNß and SC-Ad-Spike generated comparable levels of anti-SARS-CoV-2 Spike IgG in the blood, regardless of the route of vaccination (Figure 2). However, our data clearly show that the intranasal delivery of these viruses was significantly superior to intramuscular delivery for the local generation of anti-Spike CD8 T cells in the lung, where the first encounter with the SARS-CoV-2 infection is most likely to occur (Figure 2).

The data indicate that the intranasal expression of Spike from either VSV-Spike-IFNß or SC-Ad-Spike generates very high levels of anti-Spike IgG, showing that both platforms are very potent immunogens in these mice (Figure 3). Furthermore, we did not observe a significant improvement in these responses after prime and boost. This may be because the induced IgG responses are at near-maximal levels at the doses used in our experiments for the sensitivity of our assay. Thus, it may be possible to detect further increases in IgG levels after prime–boost that would only become apparent with (a) lower initial priming doses of either vector and/or (b) a longer time interval between prime and boost. However, in the K18-hAce2 murine model used here, homologous prime and boost with SC-Ad-Spike:SC-Ad-Spike was sufficient, in itself, to generate optimal IgA responses in the lungs. Sc-Ad-Spike alone was also the most effective vaccine to induce anti-Spike CD8+ T-cell responses in the lungs, which were not significantly enhanced by prime and boost regimens. Moreover, SC-Ad-Spike significantly boosted anti-Spike T cells following priming with VSV-Spike-mIFNß, although the levels of anti-Spike-specific CD8 T cells were no higher than with homologous SC-Ad-Spike:SC-Ad-Spike. The overall efficacy of SC-Ad-Spike, alone or in combination, may be attributable to the very high levels of sustained (3–7 days) Spike expression induced by this vector compared to a relatively more short-lived expression (~12 h) from the VSV platform. More detailed experiments to test this are underway.

Towards the goal of inducing anti-Spike boostable T-cell memory responses that protect against multiple SARS-CoV-2 variants, we showed that T-cell responses against Spike were induced by both the VSV or SC-Ad platforms. Importantly, however, heterologous prime and boost with VSV-Spike-mIFNß:SC-Ad-Spike was more effective for the generation of immunity in the spleen, but not in the BAL, than individual vaccinations. Thus, although heterologous prime and boost improved upon individual vaccinations systemically (Figure 3G), this was not the case for local vaccinations at the most relevant portal of the initial infection for either Ab or T-cell levels (Figure 3). Therefore, it is important to assess antibody/T-cell responses in the most appropriate location, such as where the initial infection may be occurring.

Generally, both antibody and T-cell responses were lower in older mice in response to the vaccines (Figure 4), which is consistent with the severity of SARS-CoV-2 infections increasing with age in humans. Again, SC-Ad-Spike generated the highest levels of immune reactivity (alone or in prime and boost) in these aged mice.

Most vaccinated individuals will have received both primary and boosted anti-SARS-CoV-2 vaccinations IM. Significantly, we showed here that intranasal boosting with SC-Ad-Spike following intramuscular vaccinations substantially enhanced anti-Spike IgA and CD8+ T cells compared to no boost or a VSV-Spike-mIFNß boost. Moreover, an intranasal boost notably raised both anti-Spike IgA and anti-Spike CD8+ T-cell levels compared to an intramuscular boost (Figure 5), indicating that serious consideration should be given to this route of boosting with anti-SARS-CoV-2 vaccines in the future.

We recognize several limitations to our current study. For example, the profile of expression of both the ACE2 receptor (VSV-Spike-mIFNß vaccine) and the CAR/integrins (SC-Ad-Spike vaccine) in K18-hAce2 mice will differ from the profile in humans, leading to differences in levels of infection and immunogen expression between mice and humans. Moreover, it is difficult to compare appropriate vaccinating doses of different viruses, such as the equalization of doses for in vivo administration by viral particles, infectious units, levels of induced Spike, or the maximal achievable dose. Therefore, given these variables between both experimental host and virus variables, our data here cannot be interpreted as reflecting a rank order of efficacy of the ability of the VSV-Spike-mIFNß and SC-Ad-Spike vaccine platforms to generate anti-Spike immune responses in any other host other than mice. Thus, although the SC-Ad-Spike vector appeared to outperform the VSV-Spike-mIFNß vector in the K18-hAce2 murine model, this will not necessarily be reflected in human studies. Finally, our data here do not directly address whether these antibodies and/or CD8+ T-cell responses are protective against challenges with SARS-CoV-2 or its variants, and extensive virus challenge experiments in SARS-CoV-2-susceptible hosts (such as hamsters) will be required to answer that question.

## 5. Conclusions

In summary, we showed here that both VSV and SC-Ad virus platforms can generate anti-Spike antibody and T-cell responses and that these responses fall in magnitude with age. The addition of the IFNß gene to a VSV-based vaccine significantly enhanced its safety and allowed its use for intranasal delivery. In the ACE2-transgenic mouse model used here, the SC-Ad-Spike vaccine was able to induce maximal levels of IgG, IgA, and CD8+ T-cell responses either alone or in homologous prime and boost contexts, but the true performance of the vectors will have to be assessed in humans. Our data show that intranasal boosting is significantly superior to intramuscular boosts in generating sentinel immune effectors at the site of SARS-CoV-2 entry. This indicates that serious consideration should be given to this route of boosting with anti-SARS-CoV-2 vaccines in the future.

## Figures and Tables

**Figure 1 vaccines-12-01191-f001:**
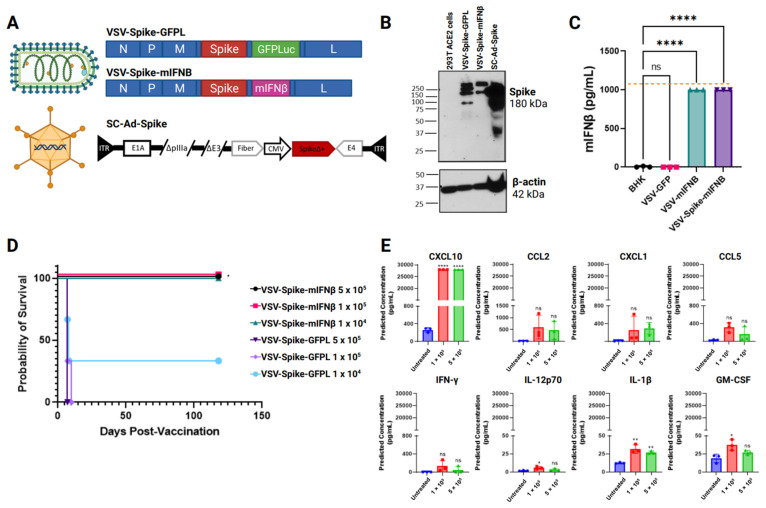
Safety and immunogenicity of intranasal delivery of viral vaccines expressing SARS-CoV-2 Spike. (**A**) Schematic of Adenoviral and VSV-vectors expressing Spike. (**B**) Detection of SARS-CoV-2 Spike protein by Western Blot 24 h post-infection of 293T ACE2 cells with different viral vaccines. (**C**) IFNβ detection by ELISA of 293T ACE2 cells infected with VSV-based vaccines. *p* < 0.0001 (****), ns = statistically non-significant, One-way ANOVA. (**D**) Survival curve of transgenic k18-hACE2 mice infected intranasally with different doses of VSV-based vaccines. *p* = 0.031 (*), Log–rank test with Grehan–Breslow–Wilcoxon test. (**E**) LegendPlexCytokine brain profile of mice experiencing toxicity 5–6 days after intranasal inoculation of VSV-Spike-GFPL.IL-12p70 *p* = 0.0333 (*), IL-1β 1 × 10^5^ *p* = 0.0016 (**), IL-1β 5 × 10^5^ *p* = 0.0068 (**), GMCSF *p* = 0.0184 (*), *p* < 0.0001 (****), One-way ANOVA.

**Figure 2 vaccines-12-01191-f002:**
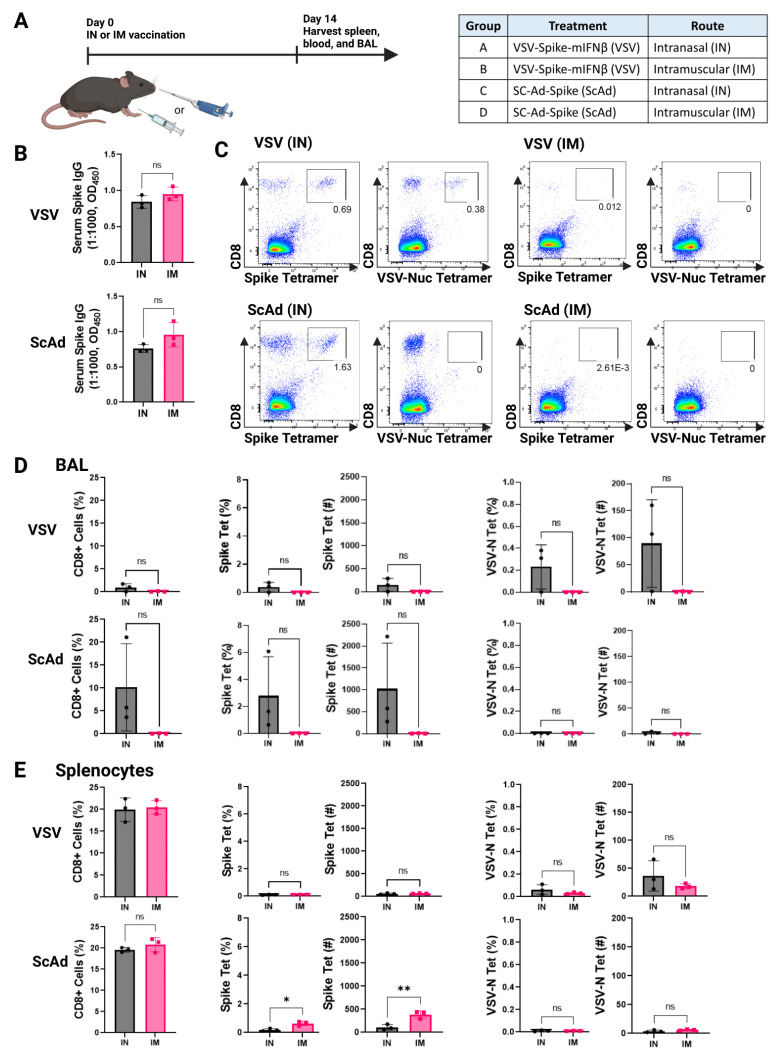
Intranasal, but not intramuscular, delivery of viruses induces T-cell responses in the lung airways. (**A**) K18-hACE2 mice were vaccinated intranasally (IN) or intramuscularly (IM) with VSV-Spike-mIFNß (dose of 2 × 10^8^ pfu) or SC-Ad-Spike (dose of 10^10^ vp). Spleens, bronchoalveolar fluid (BAL), and blood were collected 14 days post-vaccination. (**B**) Serum was used at 1:1000 dilution to test for anti-SARS-CoV-2 Spike IgG antibodies by ELISA. Similar amounts of antibody were produced IN and IM for both vaccines. *p* = ns, one-way ANOVA. (**C**) Representative flow plots of BAL samples showing anti-Spike-tetramer+ gate and VSV-Nucleocapsid tetramer+ gate. *N* = 3 mice per group. (**D**) Quantification of BAL flow cytometry results. Frequency (%) of CD8+ T cells. Frequency (%) and count of CD8+ Spike tet+ T cells, and CD8+ VSV-Nuc tet+ T cells within lymphocytes detected by flow cytometry. *p* = ns, Unpaired *T*-test. (**E**) Quantification of splenocyte flow cytometry results. Frequency (%) of CD8+ T cells. Frequency (%) and count of CD8+ Spike tet+ T cells, and CD8+ VSV-Nuc tet+ T cells within lymphocytes detected by flow cytometry. *p* = 0.0096 (**), *p* = 0.0111 (*), Unpaired *T*-test.

**Figure 3 vaccines-12-01191-f003:**
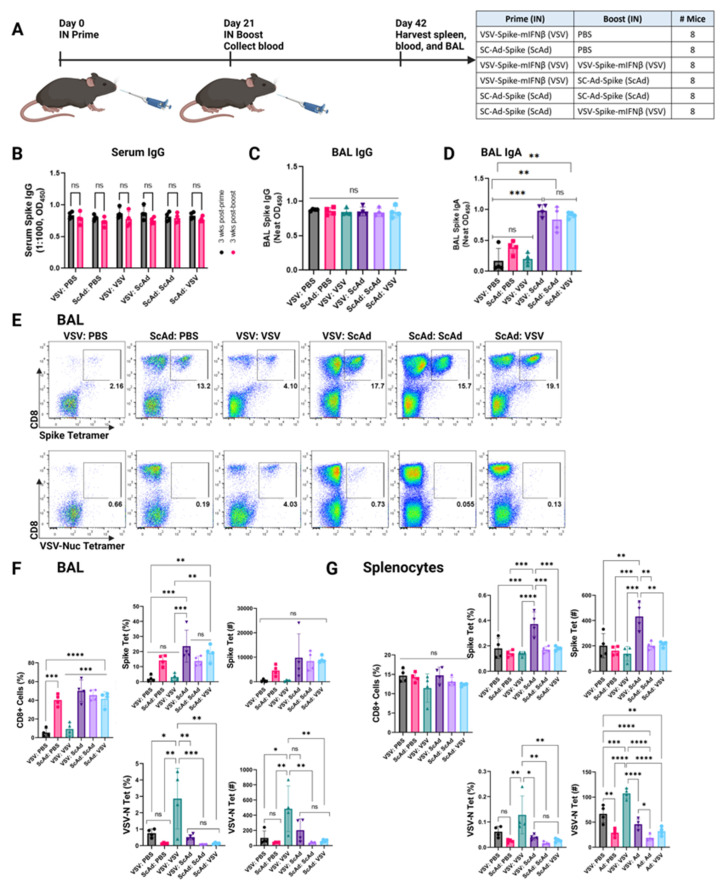
SC-Ad-Spike induces anti-Spike CD8+ T-cell responses in the lungs independent of prime and boost regimens. (**A**) K18-hACE2 mice were vaccinated intranasally (IN) with VSV-Spike- mIFNß (dose of 2 × 10^8^ pfu) or SC-Ad-Spike (dose of 10^10^ vp). Blood collected on day 21 prior to intranasal (IN) boost vaccination. Spleens, bronchoalveolar fluid (BAL), and blood were collected 3 weeks post boost. (**B**) Serum used at 1:1000 dilution to test for anti-SARS-CoV-2 Spike IgG antibodies by ELISA. Similar amounts of antibody were produced for all prime and boost combinations. *p* = ns (**C**) BAL fluid used neat to test for anti-SARS-CoV-2 Spike IgG antibodies by ELISA. *p* = ns. (**D**) BAL fluid used neat to test for anti-SARS-CoV-2 Spike IgA antibodies by ELISA. *p* = 0.0005 (***), *p* = 0.0024 (**) (**E**) Representative flow plots of BAL samples showing anti-Spike-tetramer+ gate and VSV-Nucleocapsid tetramer+ gate. *N* = 3 mice per group. (**F**) Quantification of BAL flow cytometry results. Frequency (%) of CD8+ T cells. Frequency (%) and count of CD8+ Spike tet+ T cells, and CD8+ VSV-Nuc tet+ T cells within lymphocytes detected by flow cytometry. *p* < 0.0001 (****), *p* = 0.0006 (***), *p* = 0.0068 (**), *p* = 0.012 (*), one-way ANOVA. (**G**) Quantification of splenocyte flow cytometry results. Frequency (%) of CD8+ T cells. Frequency (%) and count of CD8+ Spike tet+ T cells, and CD8+ VSV-Nuc tet+ T cells within lymphocytes detected by flow cytometry. *p* < 0.0001 (****), *p* = 0.0006 (***), *p* = 0.0068 (**), *p* = 0.0246 (*), one-way ANOVA.

**Figure 4 vaccines-12-01191-f004:**
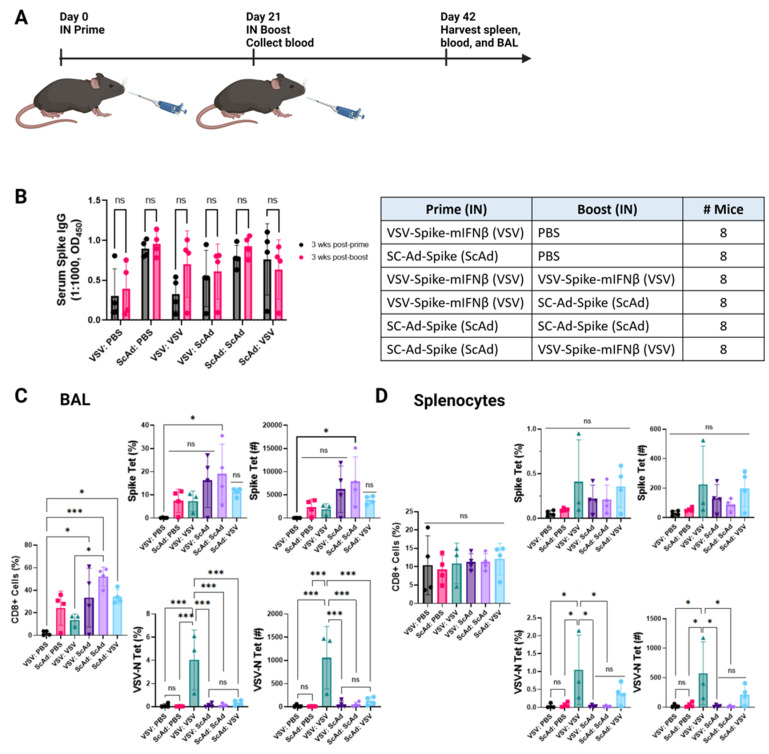
Anti-Spike Ab and T-cell response decrease with age. (**A**) Old (263 week) k18-hACE2 mice were vaccinated intranasally (IN) with VSV-Spike- mIFNß (dose of 2 × 10^8^ pfu) or SC-Ad-Spike (dose of 10^10^ vp). Blood collected on day 21 prior to intranasal (IN) boost vaccination. Spleens, bronchoalveolar fluid (BAL), and blood were collected 3 weeks post boost. (**B**) Serum used at 1:1000 dilution to test for anti-SARS-CoV-2 Spike IgG antibodies by ELISA. Similar amounts of antibody were produced for all prime and boost combinations. *p* = ns (**C**) Quantification of BAL flow cytometry results. Frequency (%) of CD8+ T cells. Frequency (%) and count of CD8+ Spike tet+ T cells, and CD8+ VSV-Nuc tet+ T cells within lymphocytes detected by flow cytometry. *p* = 0.0009 (***), *p* = 0.0438 (*), *p* = ns, one-way ANOVA. (**D**) Quantification of splenocyte flow cytometry results. Frequency (%) of CD8+ T cells. Frequency (%) and count of CD8+ Spike tet+ T cells, and CD8+ VSV-Nuc tet+ T cells within lymphocytes detected by flow cytometry. *p* = 0.0302 (*), *p* = ns, one-way ANOVA.

**Figure 5 vaccines-12-01191-f005:**
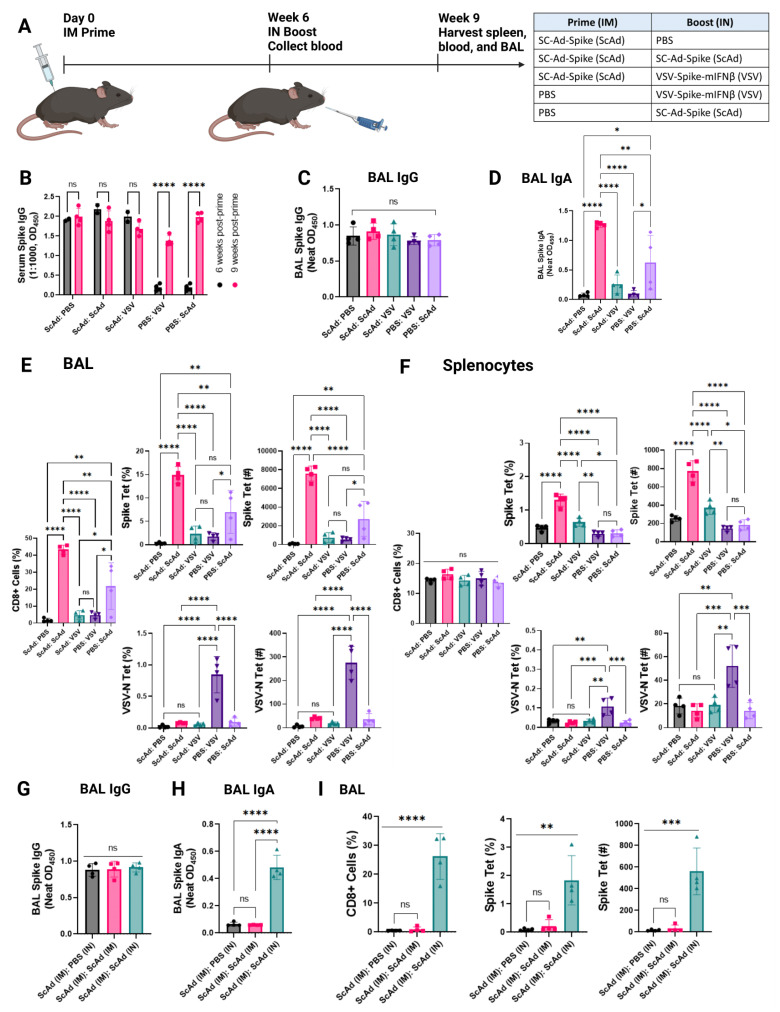
Intranasal boosting significantly enhances IgA and CD8+ T-cell responses compared to intramuscular boosting. (**A**) K18-hACE2 mice were vaccinated intramuscularly (IM) with Sc-Ad-Spike (dose of 10^10^ vp). Six weeks later, blood was collected prior to intranasal (IN) boost vaccination with VSV-Spike- mIFNß or SC-Ad-Spike or PBS. Spleens, bronchoalveolar fluid (BAL), and blood were collected 3 weeks post boost. (**B**) Serum used at 1:1000 dilution to test for anti-SARS-CoV-2 Spike IgG antibodies by ELISA. *p* < 0.0001 (****), *p* = ns, one-way ANOVA. (**C**) BAL fluid used neat to test for anti-SARS-CoV-2 Spike IgG antibodies by ELISA. *p* = ns, one-way ANOVA. (**D**) BAL fluid used neat to test for anti-SARS-CoV-2 Spike IgA antibodies by ELISA. *p* < 0.0001 (****), *p* = 0.0067 (**), *p* = 0.0292 (*), *p* = ns, one-way ANOVA. (**E**) Quantification of BAL flow cytometry results. Frequency (%) of CD8+ T cells. Frequency (%) and count of CD8+ Spike tet+ T cells, and CD8+ VSV-Nuc tet+ T cells within lymphocytes detected by flow cytometry. *p* < 0.0001 (****), *p* = 0.0094 (**), *p* = 0.0447 (*), *p* = ns, one-way ANOVA. (**F**) Quantification of splenocyte flow cytometry results. Frequency (%) of CD8+ T cells. Frequency (%) and count of CD8+ Spike tet+ T cells, and CD8+ VSV-Nuc tet+ T cells within lymphocytes detected by flow cytometry. *p* < 0.0001 (****), *p* = 0.0006 (***), *p* = 0.0087 (**), *p* = 0.0123 (*), *p* = ns, one-way ANOVA. (**G**) K18-hACE2 mice were vaccinated IN or IM with SC-Ad-Spike or VSV-Spike-mIFNβ. Blood collected on day 21 prior to intranasal boost vaccination with PBS, VSV-Spike-mIFNβ, or Sc-Ad-Spike. Spleens, bronchoalveolar fluid (BAL), and blood were collected 3 weeks post boost. BAL fluid used undiluted to test for anti-SARS-CoV-2 Spike IgG antibodies by ELISA. *p* = ns, one-way ANOVA. (**H**) BAL fluid used neat to test for anti-SARS-CoV-2 Spike IgA antibodies by ELISA. *p* < 0.0001 (****), *p* = ns, one-way ANOVA. (**I**) Quantification of BAL flow cytometry results. Frequency (%) of CD8+ T cells. Frequency (%) and count of CD8+ Spike tet+ T cells. *p* < 0.0001 (****), *p* = 0.0006 (***), *p* = 0.0044 (**), *p* = ns, one-way ANOVA.

## Data Availability

The data presented in this study are available within the document.

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
