# Peer review of "Intranasal Prime–Boost with Spike Vectors Generates Antibody and T-Cell Responses at the Site of SARS-CoV-2 Infection"

_vaccines, 2024, doi:10.3390/vaccines12101191_

Round 1
Reviewer 1 Report
Comments and Suggestions for Authors
In this work, the authors used a transgenic human ACE2 murine model to found, if a heterologous intranasally (IN) prime and boost vaccine regimen will generate improved immune responses compared to IM vaccine/booster regimens. The manuscript is well organized; however, the results presentation must be improved, also the introduction and material and methods must be improved.
1. In the Introduction, check the format of reference 6
2. Reference 12 is not suitable, it is better when the author cite other recent reference for vaccine against covide-19
3. For the paragraph „Current approved RD-Ads Covid19…. side-effect of vaccination” the author needs to add a reference.
4. There are different previous work using IN vaccination, it well be good, when the author mentions some previous vaccines using this strategy versus IM to show the expected advantages and strengths of nasal vaccination and precise, on which basis you built your hypotheses.
5. For reagents, instrument, kits, software…etc. mention the company (including city and country) from where you get it.
6. Describe the abbreviation at the first appearance, what is the full name for GFPL? The same for the other abbreviation in the text, TCID50 was first appeared in line 105
7. For the cell BHK, mention the catalog number
8. Check the reference 37 and 38???
9. About 50% of the references are not recent (published before 10 years ago), try to add recent references
10. In western Blotting, add the full name for RIPA, PVDF, TBST, ECL …etc. Add also the company (including city and country) from where you purchased all reagents (some you mention some not).
11. For the Anti-Viral Protein Assay, the analysis was done using FlowJo, are you sure? The software FlowJo is for Flow cell analysis, I think you used LEGENDplex Data Analysis Software?!! Mention the version of the software and the company from where you purchased the software.
12. For in vivo studies, add the approval Nr for animal assay.
13. In all diagram, mention the number of animals in each assay.
14. In the result part, the sentences “The SC-Ad vector…. and penton base proteins for infection” should be in material and methods not in the result part
15. Can you explain why only the VSV-Spike-GFPL was toxic?
16. You must show also the cytokine expression in mice infected with VSV-Spike-mINFβ.
17. The Figures are overloaded, especially, if there is no difference between treatment!!!???
18. In the results and discussion, you mention that VSV-Spike-mIFNß and SC-Ad-Spike generated strong anti-Spike IgG responses but prime and boost regimens don’t cause any significant improvement. This is controversial, could you explain why?
Comments on the Quality of English Language
Moderate editing of English language required.
Author Response
Comments and Suggestions for Authors
Reviewer 1
- In the Introduction, check the format of reference 6
Thank you for pointing this out, reference 6 was reformatted.
- Reference 12 is not suitable, it is better when the author cite other recent reference for vaccine against covide-19
Agreed, reference 12 was replaced with a recent review paper on SARS-CoV-2 which mentions that the most approved vaccines are replication-defective.
- For the paragraph „Current approved RD-Ads Covid19…. side-effect of vaccination” the author needs to add a reference.
We appreciate the comment, the reference was already in the paper elsewhere, but a citation has now been added to the sentences.
- There are different previous work using IN vaccination, it well be good, when the author mentions some previous vaccines using this strategy versus IM to show the expected advantages and strengths of nasal vaccination and precise, on which basis you built your hypotheses.
Two previous studies have been cited to strengthen the basis of the hypothesis. Additional background information on mucosal immunity has been added.
In terms of mucosal immunity, IgG has antigen-specific antibody activity in the lower respiratory tract and IgA prevents virus attachment to epithelial cells [24,25]. Previous studies have shown that intranasal administration of a live attenuated SCD9 SARS-CoV-2 vaccine induces higher levels of IgA than intramuscular administration [26]. Similarly, an adenovirus-vectored SARS-CoV-2 vaccine known as AdCOVID, which expressed the receptor-binding domain of the spike protein, elicited a strong humoral and cellular response when administered intranasally [27].
- For reagents, instruments, kits, software…etc. mention the company (including city and country) from where you get it.
The information has been updated in the methods section and all reagents and material have the company details provided.
- Describe the abbreviation at the first appearance, what is the full name for GFPL? The same for the other abbreviation in the text, TCID50 was first appeared in line 105
Thanks for pointing this out, the full name of GFPL now appears on line 96. The full name of TCID50 can also be found on line 135
.
- For the cell BHK, mention the catalog number
Thanks for pointing this out, the catalog number was added.
- Check the reference 37 and 38???
Thanks for catching the mistake, the references have been updated.
- About 50% of the references are not recent (published before 10 years ago), try to add recent references
We have added recent references which are relevant to the paper, the papers published <10 years ago are predominant now.
- In western Blotting, add the full name for RIPA, PVDF, TBST, ECL …etc. Add also the company (including city and country) from where you purchased all reagents (some you mention some not).
We thank the reviewer and we have added both the full name and the company with the catalog number.
- For the Anti-Viral Protein Assay, the analysis was done using FlowJo, are you sure? The software FlowJo is for Flow cell analysis, I think you used LEGENDplex Data Analysis Software?!! Mention the version of the software and the company from where you purchased the software.
We are grateful to the Reviewer for picking up our error here. As a result we have replaced the statement that we used FlowJo and have replaced it in the text with the correct description of the assay.
Data was analyzed using LEGENDplex provided software and statistical analysis was performed by one-way ANOVA.
- For in vivo studies, add the approval Nr for animal assay.
Thank you for pointing this out, the approval number is located on the Institutional Review Board Statement.
- In all diagrams, mention the number of animals in each assay.
The number of animals is stated in the methods section under In vivo studies (page 4) as "There were 8 mice per group across all the conducted in vivo experiments”.
- In the result part, the sentences “The SC-Ad vector…. and penton base proteins for infection” should be in material and methods not in the result part
This paragraph was relocated to the methods section where the vector design is addressed.
2.1.2. Viral Vector Design
We generated one SC-Ad-Spike and two VSV-Spike vectors which were replication competent, wherein SARS-CoV-2 Spike lacking its ER retention signal [31,40] was used to replace VSV-G glycoprotein gene (Fig. 1A). The SC-Ad vector expressed the Spike protein in the E4 region whereas the VSV vectors expressed either the Green Fluorescent Protein-luciferase (VSV-Spike-GFPL) or murine IFNß (VSV-Spike-mIFNß) genes upstream of the VSV L gene. Replacement of VSV-G with Spike makes VSV infection entirely contingent on binding to the ACE2 receptor of SARS-CoV-2. Whereas infection of target cells by the VSV-Spike vectors is mediated directly by interaction of the Spike glycoprotein with the ACE2 receptor, SC-Ad-Spike expresses, but does not use spike as a cell entry protein. Instead, it uses its fiber and penton base proteins for infection. SC-Ad and VSV viruses were validated for expression of the Spike gene by Western Blot following infection of 293T cells engineered to over-express ACE2 and of BHK cells, which naturally express ACE2 (Fig. 1B), and for mIFNß expression by ELISA (Fig.1C).
- Can you explain why only the VSV-Spike-GFPL was toxic?
To address the Reviewer’s question here, we have added the following text to the Discussion on page 16:
We believe that only the VSV-Spike-GFPL was toxic for a variety of reasons. In the first instance, this is a replication competent virus which spreads in vivo through binding of Spike to the ACE2 receptor, thereby generating progressively increasing amounts of virus. VSV has well documented neurotoxic properties and so IN spread of this virus at high enough titers will manifest as neurotoxicity [46]. In addition, viral spread and infection is associated with a cytokine release syndrome which has the potential to generate significant toxicity of itself [47]. Addition of the IFNß gene to the VSV-Spike-IFNß virus largely abrogated this toxicity because high levels of Type I IFNs such as IFNß are highly inhibitory to VSV replication and spread – hence restricting direct neurotoxicity of VSV and the generation of the associated cytokine storm [48]. Finally, since the ScAd-Spike is not a replication competent virus the toxicities associated with the replicative spread of the VSV-Spike would not be induced.
- You must show also the cytokine expression in mice infected with VSV-Spike-mINFβ.
We performed the cytokine analysis only for the VSV-GFP group in this experiment because we have already demonstrated that addition of the IFNß gene to VSV significantly reduces the toxicity and cytokine storm induced by VSV alone (5 REFS). Based on these studies we progressed the VSV-IFNß virus to clinical trials at the Mayo Clinic (NCT01628640). It was based upon this experience that we did not measure the cytokine profile of mice infected with VSV-IFNß. To address the Reviewer’s comment here we have, therefore, added the following text to the Results on page 6:
We and others [29,33,34,44,45] have previously shown that the addition of IFNß to VSV significantly reduces the toxicity and cytokine storm induced by VSV alone. Based on these studies, we would expect the magnitude of the cytokine profile shown in Figure 1E to be significantly reduced both quantitatively and qualitatively. Further studies will be required to identify the exact cytokines/chemokines responsible for the acute toxicity we observed with VSV-GFP (Figure 1D).
- The Figures are overloaded, especially, if there is no difference between treatment!!!???
We appreciate the Reviewer’s suggestion. We included the full complement of data/figures for completeness and because sometimes we feel that no difference in treatment can be as important as significant differences. Unless the Reviewer feels very strongly against it, we would appreciate it if we were permitted to leave the breadth of the figures as it is.
- In the results and discussion, you mention that VSV-Spike-mIFNß and SC-Ad-Spike generated strong anti-Spike IgG responses but prime and boost regimens don’t cause any significant improvement. This is controversial, could you explain why?
To address the Reviewer’s question here we have added the following text to the Discussion on page 17:
The data indicates that IN expression of Spike from either VSV-Spike-IFNß or SC-Ad-Spike generates very high levels of anti-Spike IgG, showing that both platforms are very potent immunogens in these mice (Fig. 3). Furthermore, we did not observe a significant improvement in these responses after prime and boost. This may be because the induced IgG responses are at near maximal levels at the doses used in our experiments here for the sensitivity of our assay. Thus, it may be possible to detect further increases in IgG levels after prime boost that would only become apparent with a) lower initial priming doses of either vector and/or b) a longer time interval between prime and boost.
Reviewer 2 Report
Comments and Suggestions for Authors
The manuscript details a comprehensive comparative study of COVID-19 vaccination in mice, utilizing a primary (IM) and a heterologous SC-boost vaccine regimen (Ad and VSV vectors) administered intranasally (IN) and measuring the humoral and cellular response. The thoroughness of these studies instills confidence in the validity of the findings. The figures, which are well presented, are crucial for understanding the experiments. The discussion effectively justifies the results. However, we missed including references and discussion of similar and related experiments with other infectious agents in the literature. Reference 40 seemed out of place, and the acknowledgments should have been inserted into the appropriate item. We suggest an upgrade of the English and replacement of words where they appear repeated and close (e.g., lines 63, 71, and others).
Comments on the Quality of English LanguageGood quality requires only minor adjustments that can be made alternatively during editing.
Author Response
Reviewer 2
The manuscript details a comprehensive comparative study of COVID-19 vaccination in mice, utilizing a primary (IM) and a heterologous SC-boost vaccine regimen (Ad and VSV vectors) administered intranasally (IN) and measuring the humoral and cellular response. The thoroughness of these studies instills confidence in the validity of the findings. The figures, which are well presented, are crucial for understanding the experiments. The discussion effectively justifies the results. However, we missed including references and discussion of similar and related experiments with other infectious agents in the literature. Reference 40 seemed out of place, and the acknowledgments should have been inserted into the appropriate item. We suggest an upgrade of the English and replacement of words where they appear repeated and close (e.g., lines 63, 71, and others).
We are grateful to the Reviewer for their thoughtful feedback and we have made the suggested changes.. We have also gone through the text, upgraded the vocabulary and modified the acknowledgements. We appreciate that the reviewer pointed out the relevance of reference 40 and so we have decided to cite Bio Render since we used it as an illustration tool.
To address the Reviewer’s comment on related studies, in the literature we have added the following text on lines 63-69.
In terms of mucosal immunity, IgG has antigen-specific antibody activity in the lower respiratory tract and IgA prevents virus attachment to epithelial cells [24,25]. Previous studies have shown that intranasal administration of a live attenuated SCD9 SARS-CoV-2 vaccine induces higher levels of IgA than intramuscular administration [26]. Similarly, an adenovirus-vectored SARS-CoV-2 vaccine known as AdCOVID, which expressed the receptor-binding domain of the spike protein, elicited a strong humoral and cellular response when administered intranasally [27].
Round 2
Reviewer 1 Report
Comments and Suggestions for Authors
1- Dapple Check the name of the Instrument used for Anti-Viral Protein Assay, I guess, Canto X flow cytometer is not the correct name, add the name, of the company for the instrument
2- Add the version number for LEGENDplex software
3- Add the software used for statistical analysis
4- The number of animals per group should appear in the diagram or the legends to the figures, that is the most usual way to show the number of replicates.
5- Add the company for ZE5 flow instrument and the company and version number for FlowJo
6- For the GFPL, do you have both protein (GFP and luciferase) in this construct? If yes, why? Normally one protein (GFP or Luciferase) is enough.
7- In line 280, you mean VSV-GFP or VSV-GFPL?
Comments on the Quality of English LanguageMinor editing of English language required.
Author Response
1- Dapple Check the name of the Instrument used for Anti-Viral Protein Assay, I guess, Canto X flow cytometer is not the correct name, add the name, of the company for the instrument
Thanks for pointing this out, the name has been updated and the company has been added.
Line 189: FACSCanto X SORP flow cytometer (BD Biosciences) with FACSDiva v8.0 software
2- Add the version number for LEGENDplex software
We are grateful for the Reviewer’s suggestion and we have added the version number.
Line 184: v.8
3- Add the software used for statistical analysis
Thanks for pointing this out but the software was already mentioned in the materials and methods section under Statistical Analysis section 2.12 line 258.
4- The number of animals per group should appear in the diagram or the legends to the figures, that is the most usual way to show the number of replicates.
Agreed, the number of animals has now been changed to appear on the diagrams.
5- Add the company for ZE5 flow instrument and the company and version number for FlowJo
We appreciate the Reviewer’s comment and we have added both the ZE5 company and the version number for Flow Jo.
Line 254-255: on ZE5 Cell Analyzer (Bio-Rad) with Everest™ v2.4 software and analyzed on FlowJo v.10.7.2.
6- For the GFPL, do you have both protein (GFP and luciferase) in this construct? If yes, why? Normally one protein (GFP or Luciferase) is enough.
We thank the reviewer for the insightful question, we do have both proteins in the same construct. This is very beneficial as they serve as complementary detection methods, one can look at both luminescence for in vivo experiments and fluorescence for in vitro experiments.
7- In line 280, you mean VSV-GFP or VSV-GFPL?
Thanks for pointing this out, it has been fixed and now appears as VSV-GFPL.
Lines 291-292: the exact cytokines/chemokines responsible for the acute toxicity we observed with VSV-GFPL in Figure 1D